# Nanostructured Polymeric, Liposomal and Other Materials to Control the Drug Delivery for Cardiovascular Diseases

**DOI:** 10.3390/pharmaceutics12121160

**Published:** 2020-11-28

**Authors:** Dimitrios Skourtis, Dimitra Stavroulaki, Varvara Athanasiou, Panagiota G. Fragouli, Hermis Iatrou

**Affiliations:** 1Industrial Chemistry Laboratory, Department of Chemistry, National and Kapodistrian University of Athens, Panepistimiopolis, Zografou, GR–15771 Athens, Greece; skourtisd@chem.uoa.gr (D.S.); dimistavrou@chem.uoa.gr (D.S.); bathanas@chem.uoa.gr (V.A.); 2Dyeing, Finishing, Dyestuffs and Advanced Polymers Laboratory, University of West Attica, DIDPE, 250 Thevon Street, GR–12241 Athens, Greece; pgfragouli@uniwa.gr

**Keywords:** cardiovascular, drug delivery systems, liposomes, polymeric nanoparticles, myocardial infarction

## Abstract

Cardiovascular diseases (CVDs) are the leading cause of death globally, taking an estimated 17.9 million lives each year, representing one third of global mortality. As existing therapies still have limited success, due to the inability to control the biodistribution of the currently approved drugs, the quality of life of these patients is modest. The advent of nanomedicine has brought new insights in innovative treatment strategies. For this reason, several novel nanotechnologies have been developed for both targeted and prolonged delivery of therapeutics to the cardiovascular system tο minimize side effects. In this regard, nanoparticles made of natural and/or synthetic nanomaterials, like liposomes, polymers or inorganic materials, are emerging alternatives for the encapsulation of already approved drugs to control their delivery in a targeted way. Therefore, nanomedicine has attracted the attention of the scientific community as a potential platform to deliver therapeutics to the injured heart. In this review, we discuss the current types of biomaterials that have been investigated as potential therapeutic interventions for CVDs as they open up a host of possibilities for more targeted and effective therapies, as well as minimally invasive treatments.

## 1. Introduction

Cardiovascular diseases (CVDs) encompass all pathologies of the heart or circulatory system, including coronary artery disease (CAD), myocardial infarction (MI), heart failure and peripheral vascular disease. These diseases are the primary mortality cause worldwide and significantly impact patients’ quality of life. More than 17.9 million people died from CVDs in 2016, representing above of one third of global deaths. Of these deaths, 85% are due to heart attack and stroke [1]. In Europe, 45% of all deaths are due to CVDs, accounting for 3.9 million deaths per year [2]. In the economic sector, the direct healthcare expenditure related to CVD treatment is estimated to cost the EU economy over 100 billion euros every year. Unfortunately, these figures are expected to increase over the next years due to an increase in CVD risk factors, such as obesity and diabetes, as well as an expansion of the elderly population [3]. To date, available therapies such as heart transplantation, pharmacological interventions, coronary artery bypass graft surgery, and ventricular assistant devices have significantly prolonged patients’ longevity. However, cost effective and less invasive treatments with higher efficacy are vital for the future.

In recent decades, the continuous progress of nanotechnology has led to a dramatic expansion of potential biomedical applications of nanomaterials. Nanoparticles have been widely employed in the diagnosis, imaging and treatment of many diseases, such as cancer. Nanomedicine for curing cancer is the main field of research for decades and several nanomedical products (e.g., Doxil, Abraxane) have been approved by the US FDA and the EMA for their clinical use in humans and many others are in advanced stages of clinical trials [4]. On the contrary, the field of cardiovascular nanomedicine has recently started growing and is still in its infancy. Nonetheless, an increase in research publications with promising lab-scale results has been reported, especially in recent years.

In this context, we report the efforts made for development of nanostructured biocompatible nanomaterials, both natural and synthetic, which exhibit great potential in cardiac or vascular repair and regeneration, either as a carrier for drug delivery or as an extracellular matrix substitute scaffold.

## 2. Cardiac Targeting Strategies and Nanomedicine

The major priority for a fully functional heart is a sufficient blood flow to the left ventricular. The blood which passes through the left side of the heart contains vital oxygen. When plaque builds up and hardens the arteries (atherosclerosis) that give the heart oxygen and nutrients, narrowed or clogged arteries are formed. As a consequence, the blood volume which reaches the heart is significantly reduced or blocked and this perfusion imbalance between supply and demand leads to MI [5]. After MI, the cardiac muscle is damaged and cardiomyocyte loss happens. Unfortunately, adult cardiomyocytes have poor proliferation ability and scar formation takes place, leading to myocardial hypertrophy, ventricular dilation, perivascular fibrosis and infract wall thinning [6]. For this reason, preventing the loss of cardiomyocyte in the early stages of an acute MI is necessary to achieve long-term efficacy in the treatment of coronary heart disease. If no action is taken, the heart will no longer pump adequately and heart failure will occur.

Current conventional CVDs treatment require the use of drugs in high concentration which results in toxicity to other organs. Therefore, a targeted delivery system should be used to accumulate CVD drugs selectively in the heart. Novel effective treatments have been presented by using nanomedicine. Drug delivery systems (DDSs) provide a new drug delivery avenue for the treatment of CVDs with the development of nanotechnology, demonstrating great advantages over conventional treatments. DDSs are a class of nanoconstructs, usually consisting of biocompatible polymers, lipids or inorganic compounds, that encapsulate hydrophobic or hydrophilic drugs and have the potential to increase their stability and water solubility, minimize their excretion of the body, control their biodistribution, and reduce inactivation or degradation, thereby improving their safety and effectiveness [7]. Apart from drugs, DDSs can also transfer to the site of interest cells, genes or other therapeutic molecules, administered in various ways, offering a plethora of options for the development of sufficient medication.

Contrary to DDS systems that have been developed for cancer treatment, where the drugs lead to tumor cell death, in CVDs the drugs lead to tissue regeneration and control of the proliferation of cardiomyocytes. In cancer treatment, the nanoparticles should escape the endothelium in the blood compartment to reach cancer tissue; therefore, prolonged circulation is required for the synthetic nanocarriers. In the case of CVDs, the nanoparticles delivered through the systemic circulation will eventually reach the myocardium within 20–25 min, since the human heart can pump 250 mL of blood per minute [8]. Nanoparticles must present high stability in order to withstand to the dynamic environment of the bloodstream and the pressure within the heart compartment, while they must not activate the immune system response, a significant extracellular barrier.

Extracellular biological barriers are common for all DDS systems including opsonization and subsequent sequestration by the mononuclear phagocyte system or nonspecific distribution of therapeutics to healthy organs as well as inadequate accumulation to target tissue. Additional extracellular obstacles are the hemorheological-blood vessel flow and pressure gradients, which are more pronounced in the heart compartment. Common intracellular obstacles involve cellular internalization, the escape from endosomal and lysosomal compartments and the presence of multidrug resistance proteins/pumps, which are all factors that inhibit the proper therapeutic outcome of nanomedicine [9].

Targeting is of primary importance for site specific drug delivery, because it implies selective accumulation of the cargo in the diseased tissue, with minimal effect on the healthy tissues. Currently, two strategies are employed, passive and active targeting [10]. In passive targeting, nanoparticles with certain characteristics, like predetermined size and surface charge, tend to accumulate nonspecifically in the diseased tissue, much more than they do in healthy tissue [11]. Especially in tumor tissues, polymeric drug delivery systems increase by 10 to 100-fold the concentration of the drug compared to results following administration of an aqueous solution of the pure drug. This observation can be interpreted by an increase in permeability of the tumor vessels, whose endothelial cells are poorly aligned with wide fenestrations and lack a smooth muscle lining. This phenomenon is usually referred as the enhanced permeability and retention (EPR) effect and was first explored in solid tumors [12]. The EPR effect has been observed not only in tumor growth, but also in chronic inflammation and infection cases, as well as in ischemic tissues and injured myocardial cells [13,14,15]. MI could cause many pathophysiological changes, including the enhanced permeability of endothelial cell membranes. Several angiogenic factors and vasoactive substances are upregulated after MI in ischemic tissue. This could activate and mobilize endothelial cells to form new leaky vessels, resulting in EPR-attributed nanoparticle localization. For example, vascular endothelial growth factor (VEGF), which is produced to promote angiogenesis and restore oxygen supply to the damaged tissue, is known to increase vascular permeability after MI [16]. However, this is a field of great controversy and there are many contradicting results for this targeting approach [17,18]. In contrast to the EPR effect in tumors, the post-MI EPR effect does not last long and begins to diminish after 24–48 h with partial functional recovery observed after two weeks [15]. A long period of treatment is generally required to treat and heal the heart after an ischemic episode to prevent negative left ventricular remodeling. This time window may be too short for the delivery of the necessary therapeutics through passive targeting strategies [19].

The most promising method, with interesting preliminary results, is active targeting drug delivery. In most cases, a typical nanoparticle carrier consists of a polymer or lipid matrix loaded with a certain medicine. In order to prevent rapid clearance of nanoparticles by the reticulo-endothelial system (RES), the surface of the nanoparticle is often coated with polyethylene glycol (PEG), a biocompatible, FDA-approved material, which increases circulatory half-life of the DDS. A vital step in nanocarrier fabrication is the functionalization of its surface with targeting moieties binding to the outer surface of the particle. Targeting ligands are usually peptides, antibodies, aptamers, inhibitors or small molecules that ensure specific interaction of the nanoparticle with specific molecules on the surface membrane of the targeting cells [20,21]. This is a common strategy for heart targeting, because during and after MI many receptors are overexpressed and appear on the surface of cardiac cells in response to stress, so they can be targeted by their complementary molecules. For instance, atrial natriuretic peptide (ANP) has been used as a targeting moiety for natriuretic peptide type-A receptors that are expressed on the surface of cardiomyocytes in case of ischemia [22,23]. The nanocarriers can also be labeled with radioactive isotopes or fluorescent dyes, which allows visualization of their accumulation in the diseased tissue. Therefore, innovative formulations through nanomedicine, like polymeric hydrogels, liposomic vesicles, gold or multifunctional silica nanoparticles that can go through a plethora of modifications have significant prospects to overcome physiological barriers and improve therapeutic outcomes in patients, and they will be discussed below [24].

Apart from the targeting, the endocytosis of therapeutic nanoparticles remains a major issue. The plasma membrane is a highly selective and effective barrier that protects all living cells from the surrounding environment and strongly limits the entry and exit of large macromolecules. Therefore, drug delivery systems need to overcome this barrier to intrude living cells. It is well known that nanoparticulate systems are able to enter cells, often through several endocytic pathways, like receptor-mediated endocytosis, macropinocytosis, clathrin-mediated endocytosis, caveolae-mediated endocytosis and phagocytosis. However, passive penetration of the plasma membrane may also occur as an alternative route. Particularly, during the receptor-mediated endocytosis of engineered NPs, receptors in the cell membrane freely diffuse to encounter and bind ligands decorated on NPs surface. The formation of ligand–receptor bonds provides a driving force for the membrane to wrap around NPs. Therefore, both receptor diffusion kinetics and thermodynamic driving force are of great importance to determine endocytosis efficiency. Afterwards, pinching-off of the membrane-bound NP carrier leads to the formation of an early stage endosome in cytoplasm and finally endosomal release of the NP during the late stage of endosome occurs. In this way, the cargo is released in cytosol. The properties of NPs, such as size, shape, rigidity and surface charge, could greatly affect their interactions with the surrounding environment and determine their fate during the drug delivery process. The effects of these factors in the endocytosis pathway are extensively investigated in the computational works of Ying Li et al. [25,26,27].

## 3. Types of Nanoconstructs for CVDs

Recently, the interest in developing DDS for targeting the cardiovascular system has increased exponentially. Many types of nano-based structures have been developed and can be utilized for the transport of drugs for this purpose. Liposomes, polymer–drug conjugates, polymeric micelles, nanoshells, and magnetic or hybrid carriers are some of these and all can be used for drug delivery [28]. In this section, we report the emerging research progress and results on heart-targeted nanoscale DDS (Scheme 1).

### 3.1. Liposomes

Among them, lipid-based nanoparticles are currently the most commonly used nanoconstructs for drug delivery. Many products based on liposomes have been available on the market for the past decade and some others are currently in clinical development. Liposomes are spherical structures that, most of the time, form vesicles. They are composed of a phospholipid bilayer membrane inside of which aqueous solutions can be entrapped. Their biocompatible and biodegradable composition, as well as their unique ability to encapsulate both hydrophilic and hydrophobic pharmaceutical agents, make liposomes excellent nanocarriers. Liposomes are often coated with biocompatible polymers, such as PEG, to prolong their circulation time and increase stability in aqueous environments. The polymer coating can also be modified to carry a functional group, which can be used as a surface marker [29]. Many experimental attempts to treat the infarcted heart include supplying growth factors, cytokines, drugs, and other biomolecules to the damaged cells in the infarcted tissue.

Dvir and coworkers designed a nanoparticulate system that could specifically target cardiac cells. These particles are angiotensin II type 1 (AT1) nano-liposomes that contain a targeting amino acid sequence (Gly-Gly-Gly-Gly-Asp-Arg-Val-Tyr-Ile-His-Pro-Phe) for AT1 receptor binding. This peptide was conjugated to carboxylic groups on the PEG-HSPC liposomes. AT1 receptors found to be overexpressed in heart after MI or heart failure. The AT1 particles with 142 nm size exhibited great targeting capability in vitro and in vivo. The authors first found in vitro that there was a 3-fold increase in the expression of the AT1 receptor in cardiac cells subjected to hypoxia (ventricular myocytes) compared with cells growing under normal conditions, and 83% of total amount of NPs accumulated in these cells. Furthermore, in in vivo experiments, fluorescent liposomes were injected in mice 1, 4 and 7 days after MI and it was found that NPs amassed predominantly in the left ventricle of the infarcted heart. This finding shows the ability of these particles to specifically target the injured myocardium and may be exploited to deliver therapeutic agents to it [30].

Yu et al., reported the fabrication of oleate adenosine prodrug lipid nanocarriers functionalized with atrial natriuretic peptide (ANP) for the treatment of MI. Adenosine is an endogenous nucleoside that regulates the proper function of the heart and brain, but its short plasma half-life hindered its clinical use. In this study, this nucleoside was conjugated to oleate acid in order to achieve better stability and sustained release. In addition, ANP, which has cardioprotective properties and its receptors are overexpressed in the endocardium of the ischemic heart [31], was conjugated to polyethylene glycol-distearoylphosphatidylethanolamine (PEG-DSPE) to get ANP-PEG-DSPE. Afterwards, lipid nanocarriers were formed via the solvent evaporation method incorporating adenosine oleate prodrug inside of their core. In vitro drug release and cell uptake studies showed that 80% of the drug was released within 48 h and high cellular uptake efficiency was observed without cytotoxic effects. In vivo experiments indicated that the infarct size was significantly inhibited and nanoparticles accumulated predominantly in the infarcted myocardium. Therefore, ANP-modified oleate adenosine lipid nanocarriers are candidates for novel drug delivery to the heart in a receptor-dependent manner due to their therapeutic efficiency and targeting ability [23].

Takahama and colleagues also presented a study in which they investigated whether liposomal adenosine has fewer side effects and offers better cardioprotection than free adenosine. For this purpose, they synthesized liposomes consisting of DSPE-PEG, cholesterol, hydrogenated soy phosphatidyl choline (HSPC) and adenosine via the hydration method. The PEGylated liposomes with a diameter of 134 nm accumulated specifically in ischemic myocardium of rats at 3 h after MI, taking advantage of the enhanced permeability effect of the affected tissue. Intravenous infusion of free adenosine led to a decrease in blood pressure and heart rate in a dose dependent manner, while liposomal adenosine did not. Radioisotope-labeled adenosine encapsulated in liposomes exhibited prolonged circulation time in comparison with the free form. Moreover, liposomal adenosine significantly reduced MI size showing increasing cardioprotective effects [32].

Dong et al. reported the synthesis of a novel solid lipid nanoparticle (SLN) system, as a transferring agent of Puerarin (7,4′-dihydroxyisoflavone-8b-glucopyranoside, PUE) for MI treatment. This SLN system consisted of two different parts and was fabricated following the solvent evaporation method. The solid core was consisted of Compritol® 888 ATO and soya lecithin and the PEG-DSPE compartment modified with cyclic arginyl-glycyl-aspartic acid (RGD) peptide was exploited for stabilization and targeting purposes. The RGD peptide was used as targeting moiety for ανβ3 integrin, which is overexpressed on endothelial cells. The size of these nanoparticles (RGD/PEG-PUE-SLN) loaded with PUE was about 110 nm, and the zeta potential was negative (−26 mV), ensuring the inhibition of platelet activation and the major protection of the heart. The encapsulation efficiency was about 85.7% for RGD/PEG-PUE-SLN and the loading capacity was 16.5%. In vitro tests showed no cytotoxicity of the SLNs and rapid drug release was observed at the first 12 h, followed by a sustained release in 48 h. In vivo studies revealed that the RGD/PEG-PUE-SLN accumulated mainly in the infarcted heart and PUE concentration was 23-fold higher than free PUE. RGD/PEG-PUE-SLN also minimized the infarct size to 6.2%, indicating the potential of this system for MI treatment [33].

Another target-specific lipid nanoparticle was reported by Dasa for the delivery of small molecule drugs after MI. This team synthesized liposomes with different targeting 7-mer peptides (obtained by phage display biopanning) with affinity for cell types present in heart after MI. The most effective system was a I-1 peptide functionalized liposome consisting of dioleoylphosphatidylcholine (DOPC) cholesterol and DSPE-PEG, 100–120 nm in size, that showed specific interaction with cardiomyocytes, but not exclusively. These liposomes were used to deliver in mice AZ7379, a PARP-1 (Poly(ADP-ribose) polymerase 1) inhibitor. PARP-1 is a nuclear enzyme that is activated in response to oxidative stress and catalyzes the formation of poly (ADP-ribose). The cardiomyocyte-targeted delivery of AZ7379 promotes cardiomyocyte-specific inhibition of PARP-1, with 9-fold higher efficiency 24 h post injection than negative control liposomes. Free AZ7379 showed very limited inhibition [34].

Wang and his group recently investigated the development of a dual modified liposome carrier with improved heart delivery capability. The targeting ligands on the surface of the NPs were two peptides, PCM (WLSEAGPVVTVRALRGTGSW) and TAT (YGRKKRRQRRR). PCM is a 20-mer peptide with the ability to bind primary cardiomyocytes with high affinity and TAT is an 11-mer peptide that can promote the intracellular penetration through nonspecific endocytosis. The liposomes composed of soybean phospholipids, cholesterol and DSPE-PEG had a size slightly over 100 nm, negative zeta potential (−10 mV) and were stable for at least 30 days in vitro. These particles showed no cytotoxic effects at a concentration bellow 0.8 mM and in vitro and in vivo experiments with fluorescent coumarin-6 exhibited high cellular internalization. Thus, this novel dual-modified delivery system provides better cardiomyocytes targeting ability and it would be able to deliver therapeutic agents to the heart effectively [35].

A similar approach was followed by Ko for gene delivery to the ischemic heart by double-targeted lipoplexes. In this case, TAT peptide and anti-myosin monoclonal antibody (mAb 2G4) were employed as targeting moieties. Anti-myosin mAb 2G4 is a highly specific antibody that has the ability to recognize and bind ischemic cells with damaged plasma membranes when intracellular myosin becomes exposed to the extracellular surface and thus allows for the antibody binding. TAT- and TAT-mAb 2G4 lipoplexes were fabricated with the addition of a cationic lipid (DOTAP) in a very small ratio to the other components, in order to complex with a negative charged plasmid DNA. Both lipoplexes exhibited in vitro transfection to rat hypoxic cardiomyocytes, with double modified NPs showing enhanced transfection and four-fold higher green fluorescence protein (GFP) expression. However, no synergistic effect was observed for the two ligands. In vivo, fluorescent polyplexes administered in an MI rat model showed preferential accumulation in the ischemic area of the heart and expression of the GFP gene, making this system a candidate for the potential delivery of therapeutic genes to the ischemic myocardium [36].

The targeting efficiency of mAb 2G4 for the cardiac myosin was exploited by Liang and coworkers for the synthesis of ATP-containing immunoliposomes. In the ischemic heart, myocardial adenosine triphosphate (ATP) utilization is reduced due to inadequate contractile function, while the need for ATP remains high. This imbalance between supply and demand for ATP deteriorates left ventricular cardiac function. Free ATP cannot be administered because of its short circulation half-life and, because of its strong negatively charged nature, cannot enter into cells through their membrane. So, in this study, ATP was encapsulated in liposomes containing phosphatidylcholine, cholesterol, PEG-DSPE and 2G4-PEG-PE. Liposomes were produced via the freeze-thawing method and after ATP entrapment only minor ATP leakage was observed after 24 h at 37 °C. The specific binding of 2G4-immunoliposomes to myosin was evaluated using ELISA. The results showed strong binding to the myosin monolayer [37]. In vivo experiments using these NPs with or without the targeting antibody were conducted by Verma et al. in rabbits after an acute myocardial infarction. Infusion of ATP-NPs followed by 30 min of coronary artery occlusion and 3 h of reperfusion, resulted in a significant decrease (over 50%) in the size of irreversibly damaged myocardium within the total area at risk. These NPs accumulated in the heart due to the EPR effect and seem to have a cardioprotective effect by supplying the myocardium with the required energy [38]. Some myosin specific liposomes demonstrated even better cardioprotection by significantly restoring the systolic and diastolic function of the heart. At the end of reperfusion, the left ventricular pressure recovered to more than 80%, while the left ventricular end diastolic pressure reduced considerably [39]. In addition, the delivery of Coenzyme Q10 (CoQ10) with similar liposomes was investigated by the same team. Intracoronary infusion of CoQ10 loaded liposomes in rabbits with MI resulted in a 50% reduction in infarct size, which could in turn be sufficient for potential recovery from an infarction in a clinical situation [40].

Scott and coworkers focused their efforts on developing anti-P-selectin conjugated lipid nanocarriers to treat rat MI. Anti-P-selectin was used as targeting ligand because it has been reported that P-selectin is upregulated in endothelial cells after ischemic or inflammatory events in several tissues. In particular, P-selectin is expressed in the region between the healthy and necrotic tissue (border zone) in infarcted heart as observed by immunohistochemical analysis. Liposomes were fabricated using HSPC, cholesterol and DSPE-PEG and anti-P-selectin was conjugated to the surface of the nanocarriers by thiol group modifications. Anti-P-selectin coated NPs were administered instantly or 4 h after MI. The immediately infused liposomes showed a significant increase in accumulation (83%) to the infarct region as compared to the non-infarcted area after 24 h circulation, while the 4 h post-MI infused liposomes showed a 92% increase in adhesion. Exploiting this property, vascular endothelial growth factor (VEGF), a proangiogenic compound, was encapsulated to the liposomes in an attempt to revascularize the infarcted myocardium. Administration of a single dose of 0.12 μg kg^−1^ targeted VEGF immunoliposomes resulted in significant improvements in cardiac function and vascular structure up to 4 weeks after MI. VEGF enhanced systolic function and fractional shortening of the infarcted tissue, while a raise in number of anatomical and perfused vessels in myocardium border zone was noticed. At this point, is very important to note that in many VEGF therapies initiated many days after MI, no therapeutic effects were detected. Therefore, the immediate exposure of the heart tissue to VEGF, which could delay the myocardial apoptosis and cardiac remodeling is of primary importance [41,42].

In addition, an interesting study was reported by Liu for the therapy of arrhythmia to the ischemic heart. They synthesized antibody modified liposomes for the targeted delivery of anti-miR-1 antisense oligonucleotides AMO-1 to the ischemic myocardium. It is well established that microRNA-1 (miR-1) can be found in cardiac and skeletal muscle tissues and is overexpressed in the ischemic heart. The suppression of miR-1 by AMO-1, an anti-miR-1 antisense oligonucleotide, can lead to the therapy of cardiac arrhythmia. Furthermore, cardiac troponin I (cTn-I) is a specific marker which is expressed only in damaged heart tissues. Based on this, anti-cTn-I antibody was harnessed as a targeting moiety. The liposomes, containing egg phosphatidylcholine, cholesterol and DSPE-PEG, were fabricated via the thin film hydration method, and the antibody was conjugated using DSPE-PEG-MAL. The NPs had a slightly negative surface charge, a size close to 100 nm, high AMO-1 entrapment efficiency (>60%) and 23% in vitro release rate in 24 h. Time lapse live cell imaging and in vivo imaging utilized to monitor anti-cTn-I NPs and both demonstrated high accumulation in the heart and specifically in the cytoplasm of myocardial cells. After administration of AMO-I loaded NPs in rats, electrocardiograph recordings exhibited repression of arrhythmogenesis. To conclude, anti-cTnI liposomes not only delivered AMO-1 to infarcted myocardium in rats, but also relieved cardiac arrhythmia via the silencing of miR-1 in the ischemic heart [43]. All efforts presented above are summarized in Table 1.

### 3.2. Polymeric Nanoparticles

Biodegradable polymeric nanoconstructs have been extensively investigated as drug carriers. This type of nanocarrier includes copolymers consisting of two or more blocks with different hydrophobicity. In aqueous environments, these copolymers usually form micellar structures composed of a hydrophilic shell and a hydrophobic core. Inside the hydrophobic core, water-insoluble drugs can be hosted, while the hydrophilic corona can be additionally modified for the attachment of water-soluble drugs or targeting moieties. [44]. However, this is not always the case. To date, many other polymeric nanoconstructs have been synthesized with very different structures. For instance, polymers can form hydrogels [45,46,47], network-like scaffolds [48], microparticles [49], nanospheres or nanoshells that are suitable not only for the release of hydrophobic but also hydrophilic drugs, proteins, genes, nucleic acids or genes. [50]. The release of encapsulated drugs occurs at a controlled rate in a time or environment dependent manner. The kinetics of the release of nanoparticle-bound drugs depend on the duration of nanoparticles residence in the organism and characteristics of their microenvironment. In general, biodegradable polymer formulations maintain the desired therapeutic drug concentration in the tissue of interest for a longer period of time than other nanocarriers. This feature makes them suitable platforms for the transport of highly toxic, poorly soluble, and unstable drugs.

Poly(d,l-lactic acid), poly(d,l-glycolic acid) (PLGA) is a copolymer that has been widely used for medical applications due to its low toxicity, high biocompatibility and biodegradability and because it is an FDA approved material. In this route, many research groups have focused their attempts on developing PLGA based nanomedicine materials. For example, Chang and coworkers presented the synthesis of insulin-like growth factor-1 (IGF-1) complexed PLGA NPs for early cardioprotection after acute MI. IGF-1 is crucial in regulating heart function and it has been found to promote cardiomyocyte proliferation. In humans, IGF-1 treatment enhances cardiac function after infarction. In order to maintain its biological function, anionic IGF-1 formed complexes with PLGA via electrostatic interactions. Because PLGA is also negatively charged, polyethylenimine conjugation was utilized to render PLGA’s surface cationic and to achieve PLGA-IGF-1 complexation. Various-sized NPs were synthesized (1 μm, 200 nm, 70 nm), with the smallest ones (70 nm) found to have the best binding capability and activated the most Akt phosphorylation in cultured cardiomyocytes. After a single intramyocardial injection in mice, IGF-1-PLGA NPs prolonged IGF-1 retention time to at least 24 h, inhibited cardiomyocyte apoptosis, reduced infarct size and prevented ventricular dilation and wall thinning 21 days after MI. These results highlight the potential of nanomedicine in treating cardiovascular diseases and encourage its translation into clinical applications [51].

Administration of statins at the time of reperfusion exhibited no therapeutic effects in animals and in patients with acute MI. Nevertheless, Nagaoka and his team found that nanoparticle delivered Pitavastatin presented cardioprotective effects. They synthesized Pitavastatin-PLGA NPs that accumulated in the infarcted region of the myocardium of a rat MI model due to its enhanced vascular permeability. After single intravenous injection at reperfusion, Pitavastatin NPs reduced the size of infarct after 24 h and improved left ventricular function. They also induced phosphorylation of Akt and GSK3β and inhibited inflammation and cardiomyocyte necrosis in the infarcted heart, while the free Pitavastatin drug failed to show any cardioprotection. Moreover, Pitavastatin NPs have successfully completed phase I clinical trials at a university hospital, suggesting that this nanoparticle-based technology can serve as novel therapeutic treatment for ischemic injury [52].

In coronary artery disease (CAD) and peripheral artery disease (PAD), therapeutic angiogenesis has appeared as a potential clinical approach for the recovery of ischemic tissues, including the direct administration of pro-angiogenic growth factors to promote the synthesis of growth factors by target tissues. In this context, adrenomedullin-2 (ADM-2) has been recently identified as a new angiogenic factor able to regulate blood flow and cardiovascular function. Due to its short biological half-life and low plasma stability, ADM-2 has, up to now, limited applications. For this reason, Quadros and coworkers tried to encapsulate this peptide into PLGA nanoformulations that could serve as delivery system for heart repair. PLGA-ADM-2 NPs, prepared by a double emulsion method, had a size of 300–350 nm, negative surface charge, good colloidal stability and did not present toxicity in cardiomyocyte cells. The entrapment efficiency of ADM-2 reached circa 70%, as identified by ELISA, and in vitro peptide release showed a 60% release in the first 3 days followed by a slower release rate in the next 3 weeks. In vitro experiments in EA.hy926 endothelial cells also exhibited enhanced cell proliferation (1.4 fold higher), rendering AMD-2-PLGA NPs a novel approach for therapeutic angiogenesis in CVDs [53].

In a pioneering work, Nguyen et al. presented a system for potential treatment of heart failure following a MI based on intravenous administration of enzyme responsive nanoparticles. These nanomaterials have the ability to respond to the enzymatic conditions present in the heart after an acute MI by changing their structure from discrete micellar nanostructures to network-like scaffolds. It is worth noting that these nanosystems (Figure 1), consisting of brush peptide–polymer amphiphiles based on a polynorbornene backbone with peptide targeting sequences specific for recognition by matrix metalloproteinases MMP-2 and MMP-9, have the ability to remain stable during blood circulation conditions until they reach the infarct region through the vascular leakage after MI. At this site, their structure is altered, and it has been found that they remain on the injured tissue for up to 28 days after injection. In previous work, super resolution fluorescence microscopy measurements showed that the responsive nanoparticles are MMP-9 activated and they are kept assembled into scaffolds at the injection site for 7 days, while the non-responsive particles were inactive in the presence of MMP-9 and were cleared within 1 h. Furthermore, the efficacy of the system was also studied in rats with MI and it was indicated that the responding particles were activated in the presence of MMPs and remained aggregated in network-like scaffolds for 6 days, in contrast with the reduced aggregation observed in non-responsive particles. Finally, after injecting responsive and non-responsive particles into the tail vein of healthy rats as well as in 24 h post-MI rats, a targeted and prolonged accumulation of responsive particles in the diseased heart tissue was demonstrated for 28 days. In conclusion, MMP-responsive nanoparticles exploit the EPR effect for initial passive targeting and present a very promising approach to deliver therapeutics to the heart after MI in an effective and controlled manner [48].

Furthermore, for the promotion of angiogenesis in the infarcted tissue, Garbern and her team designed a pH- and temperature responsive nanocarrier for the delivery of basic fibroblast growth factor (bFGF). They used the random copolymer poly(*N*-isopropylacrylamide-*co*-propylacrylic acid-*co*-butyl acrylate) (p[*N*IPAAm-*co*-PAA-*co*-BA]), synthesized by reversible addition fragmentation chain transfer (RAFT) polymerization, because of its ability to undergo a sol–gel phase transition. This polymer is liquid at pH = 7.4 and becomes a gel at pH = 6.8, exploiting the lower pH of the ischemic region. In vivo experiments in rat hearts showed that the injectable hydrogel enhanced local accumulation of bFGF in the apex of the heart, with minor diffusion to other regions of the tissue. In addition, the retention of bFGF 7 days post injection was 10-fold higher with hydrogel than with saline delivery, and, after 28 days of treatment, the relative blood flow was significantly increased (2-fold increase) as compared to day 0. In addition, capillary and arteriolar densities were increased by over 30%, while fractional shortening, determined by echocardiography, was remarkably increased following treatment with this polymeric hydrogel [46].

A different approach was followed by Sy for the inhibition of cardiac dysfunction after MI. They presented the synthesis of poly(cyclohexane-1,4-diylacetone dimethylene ketal) (PCADK) microparticles and the encapsulation of SB239063, a p-38 inhibitor, into these particles. The p38 mitogen-activated protein kinase pathway plays a key role in activating macrophages and in inducing cardiomyocyte apoptosis. For this reason, PCADK-p38 particles of 10–20 μm size formed via a single emulsion/evaporation method and the encapsulation efficiency of the inhibitor reached nearly 50% with a concentration in microparticles of 3–5 μg of inhibitor per mg of polymer. In vitro experiments in cultured macrophages showed that PCADK-p38 microparticles prevented p-38 phosphorylation by TNF-a stimulation, reduced superoxide production (an inflammatory effector) and at the same time an accelerated release of the inhibitor was observed, possibly due to intracellular release after phagocytosis. In addition, PCADK-p38 microparticles did not induce any inflammatory response in vitro and in vivo. After a single intramyocardial injection in mice, the microparticles remained in the heart for days and seriously inhibited p-38 phosphorylation within the infarct zone, whereas free SB239063 had no effect. At 21 days postinfarction, an obvious improvement in fractional shortening and a reduction in fibrosis were observed [49]. A similar study was presented by Gray for the development of smaller (<500 nm) *N*-acetylglucosamine functionalized PCADK nanoparticles. These NPs carried the same inhibitor, but *N*-acetylglucosamine on their surface promoted cell internalization. In vitro and in vivo experiments also showed promising results, as administration of PCADK NPs reduced apoptotic events and infarct size and improved acute cardiac function [54].

Somasuntharam and coworkers developed a nanocarrier for the delivery of Nox2-siRNA to the infarcted heart environment through particles based on acid-sensitive polyketals, in order to silence the Nox2 gene in cardiac macrophages. Nox2 is a catalytic subunit for nicotinamide adenine dinucleotide phosphate (NADPH) oxidase, responsible for reactive oxygen species production, and Nox2-NADPH upregulation mainly occurred in the infarcted myocardium. Silencing of this gene has been found to have cardioprotective properties. So, polyketal PK3 copolymer was used for the incorporation of Nox2-siRNA in the nanoparticles. The encapsulation efficiency of siRNA within the PK3 particles was found to be 40%, while their average size was 500 nm. The internalization of siNox2 loaded particles by macrophages was found to be over 80%, as determined by flow cytometry. Further experiments in the mouse macrophage cell line RAW 264.7 also confirmed that PK-siNox2 particles suspended Nox2 mRNA expression by over 40%. In vivo experiments in mice also showed lower levels of Nox2 mRNA expression and echocardiography data evidenced that cardiac function improved significantly after NPs intramyocardial administration. This promising Nox2-siRNA delivery system is stable during blood circulation, leading to a reduction in oxidative stress and protection of the heart from failure [55].

Huang et al. reported the synthesis of cysteine–arginine–glutamic acid–lysine–alanine (CREKA)-modified NPs for the delivery of thymosin beta 4 (Tβ4) to the infarcted myocardium. Tβ4 is a 43 amino acid, expressed in the embryonic heart, responsible for epicardial and coronary artery growth and overexpressed in the infarcted heart. Recent research has uncovered that Tβ4 enhances heart function by protecting cardiomyocytes from apoptosis and by activating epicardial progenitor cells. However, free Tβ4 administration is unfeasible due to short circulation half-life. Fibrin is predominantly expressed in the early stages of MI in the injured area of the heart and constitutes a potential target for DDS. CREKA, a fibrin targeted peptide and Tβ4 were both conjugated to the outer area of poly(ethylene glycol)–poly(lactic acid) (PEG-PLA) NPs via EDC/NHS and maleimide–thiol reactions, respectively. The NPs produced by an emulsion/solvent evaporation protocol had a size of 100 nm and negative zeta potential (−10 mV). Ex vivo heart imaging experiments revealed that CREKA-NPs accumulated in the infarcted area, and this mostly occurred 24 h after administration. Interestingly, bare NPs (without CREKA) also accumulated in the same area because of the EPR effect. Tail vein injection of CREKA-NPs in mice showed multiple therapeutic effects, such as reduction in end diastolic and systolic diameter, 30% reduction in infarct size, increased wall thickness, increased vessel and arteriole density and cell proliferation in the peri-infarct area. All these results indicate that this delivery system is very promising for further clinical applications [56].

Diabetic cardiomyopathy (DCM) is a CVD caused by diabetes and it has many adverse effects in heart function, like systolic/diastolic cardiac changes and atherosclerotic heart disease that later leads to heart failure or ischemia. To tackle DCM, Tong and coworkers developed curcumin polymeric carriers consisting of a triblock copolymer poly(γ-benzyl l-glutamate)-poly(ethylene glycol)-poly(γ-benzyl l-glutamate) (PBLG-PEG-PBLG) (P). The copolymer was synthesized by the ring opening polymerization of the BLG-N-carboxy anhydride with H_2_N-PEG-NH_2_ as the macroinitiator. The curcumin incorporated particles had a diameter of 30 nm and the loading capacity of curcumin was over 30%. These particles showed very limited cell toxicity in vitro in high concentrations and curcumin release was achieved within 3 days. In previous studies, curcumin was found to have therapeutic effects on diabetes and to alleviate DCM. Indeed, it was found that curcumin loaded NPs reduced pathological morphological damage of cardiac tissue and increased H_2_S levels that play a vital role in the inhibition of apoptosis in cardiomyocytes [57].

Hardy et al. designed a double loaded drug delivery system against l-type Ca^2+^ channel. The l-type Ca^2+^ is the main source for calcium in cardiomyocytes. When ischemia reperfusion injury occurs, increasing levels of intracellular calcium and reactive oxygen species are the two main factors contributing to cardiac hypertrophy, inflammation and cell apoptosis. To deal with this, they designed poly(glycidyl methacrylate) (PGMA) nanoparticles loaded either with curcumin or resveratrol and a peptide derived against the alpha-interacting domain of the L-type Ca^2+^ channel (AID), with amino acid sequence QQLEEDLKGYLDWITQAE, that has been found to reduce heart injury post reperfusion. The NPs constructed through oil in water emulsion process and AID peptide were electrostatically attached to the surface of NPs after modification of PGMA with positively charged polyethylenimine. The loading efficiency of resveratrol was quite low (1% *w*/*w*) with fast release rate compared to curcumin and AID loading that reached 12% *w*/*w* with a sustained release profile within 60 min of reperfusion. The curcumin-AID NPs were found to be more effective in minimizing oxidative stress and superoxide production ex vivo in rat hearts, while resveratrol NPs were found to have only an initial therapeutic effect, preventing their further application as potential DDS. Importantly, AID in combination with curcumin proved to ameliorate ischemia reperfusion injury via different mechanisms, rendering them as vital components of an effective DDS [58].

In 2010, Nam et al., reported the synthesis of a primary cardiomyocyte specific peptide-functionalized poly (cystamine bisacrylamide-diaminohexane, CBA-DAH) (PCD) polymer in order to deliver Fas siRNA selectively and efficiently in cardiomyocytes. The delivery of Fas siRNA leads to the silence of Fas gene expression and inhibits cardiomyocyte apoptosis. This cardiomyocyte-targeted Fas siRNA delivery system presented not only high transfection efficiency, but also low cytotoxicity. PCM peptide indicated high selectivity for cardiomyocytes and z potential measurements showed that the binding of PCM to the polymer, via amino groups, led to a reduction in cationic charges, impairing their ability to incorporate negatively charged genes. Therefore, it was necessary to introduce unmodified PCD, in a ratio of 1:1 *w*/*w* with the PCM-PCD, in order to increase the positive charge and obtain stable nanoparticles with a diameter of less than 200 nm. These polymers can form complexes with polyanionic genes efficiently, like Fas siRNA. Transfection experiments and cellular uptake studies confirmed that the biodegradable PCM-modified polymer could bind specifically to H9C2 cells, revealing the cardiomyocyte-targeting ability of PCM-PCD nanoparticles. Flow cytometry measurements in H9C2 cells indicated the efficiency of Fas gene silencing by PCM-polymer/Fas siRNA complexes to inhibit apoptosis under hypoxic conditions. Quantitative real-time PCR analysis showed a 77% reduction in Fas gene expression in H9C2 cells treated with Fas siRNA, delivered using the PCM-PCD/PCD carrier. The PCM-modified polymer/Fas siRNA polyplexes facilitate Fas gene silencing and increase cardiomyocyte viability under hypoxic conditions. To conclude, PCM modification is a promising method for cardiomyocyte targeting, and siRNA delivery with PCM-PCD polymers inhibits cardiomyocyte apoptosis [59].

Another interesting aspect for treating MI is the restoration of mechanical and electrical function of the heart by exploiting the conductive properties of some nanoformulations. As an example, a soft conductive hydrogel was presented by Bao and his group as an electrical signal transmitter for the infarcted myocardium (Figure 2). They synthesized an injectable hydrogel consisting of a multi-armed PEGDA700-Melamine (PEG-MEL) crosslinker, stabilized through π–π interactions between the triazine rings of melamine, that was used to conjugate a thiol modified hyaluronic acid (HA). Moreover, the addition of graphene oxide (GO) has provided soft (G′ = 25 Pa) and conductive (G = 2.84 × 10^−4^ S cm^−1^) properties and enhanced mechanical stress stability. To improve the healing ability of the hydrogel, adipose tissue-derived stromal cells (ADSCs) were incorporated. After an injection in the MI area in a rat model, an overall enhancement of cardiac function was observed. In particular, the ejection fraction (EF) and fractional shortening (FS) greatly improved in a one month period (EF = 78% and FS = 53%), the size of infarction and fibrosis reduced and the vascular density increased. Eventually, the overexpression of Connexin 43 and α-Smooth Muscle Actin confirmed the efficient transmission of electrical and mechanical signals to the heart [45].

The insufficient propagation of electrical signals to the heart after MI can also cause cardiac arrhythmia. For this reason, Zhang and coworkers designed an injectable conductive hydrogel that ameliorates arrhythmia and preserves cardiac function (Figure 3). This group synthesized a conductive poly-3-amino-4-methoxybenzoic acid-gelatin (PAMB-G) copolymer through an oxidative polymerization method. The PAMB is the conductive ingredient, while gelatin improves the biocompatibility of the whole system. The copolymer formed a hydrogel after crosslinking of the polymer chains by using EDC/NHS chemistry. An injection to the scar zone of the heart in a rat model one week after MI, showed that the hydrogel enhanced electrical impulse transmission and activated isolated contracting regions. In addition, four weeks after treatment, regional electrical field potential amplitude around the infarcted area was greatly increased and electrocardiograph recordings indicated that fewer premature ventricular contractions occurred. Except for that, PAMB-G treated hearts presented higher fractional shortening and ejection fraction, lower left ventricular internal systolic and diastolic dimensions and a smaller fibrotic area, all resulting in cardiac function enhancement [47]. In Table 2 a synopsis of the polymeric nanostructures discussed in this section is presented.

### 3.3. Other Types

Apart from polymeric and lipid NPs, other types of nanoconstructs have also been developed for drug delivery to the myocardium. Some of them may contain inorganic components such as silica, gold or carbon nanocapsules, and others may contain biomaterials, such as alginate salts or chitosan.

Silica nanoparticles can be used for heart-targeted drug delivery in both passive and active ways. In particular, Galagudza and coworkers designed adenosine loaded silica nanoparticles for the passive delivery of adenosine to the ischemic-reperfused cardiac tissue. As mentioned before, adenosine is a cardioprotective agent and its incorporation to the NPs was achieved through adsorption on their surface. The NPs mean diameter was 6–13 nm, presenting no toxic effects in vitro and in vivo. However, their long-term toxicity remains unclear due to their capture by the liver. After intravenous administration of adenosine-silica NPs in rats with ischemic-reperfusion injured hearts, augmented accumulation in the infarcted area was observed. Adenosine bounded to NPs restored blood pressure and notably reduced the infarct size in comparison with free adenosine administration. It must be emphasized that in order to maximize the therapeutic effects of the DDS, it must be administered right after ischemia or at the beginning of reperfusion [60].

In contrast, Ferreira and her team developed multifunctional silica NPs loaded with a cardioprotective compound for active targeting delivery to the endocardial layer of the injured heart (Figure 4). The synthesized bare silica NPs with initial size of 165 nm were further functionalized with PEG and ANP peptide to reach a final size of 200 nm. The zeta potential of the NPs after ANP modification turned from negative (−30 mV) to positive (>25 mV) due to positively charged amines in the peptide sequence, thus reducing the stealth properties of the system that contribute to the possible formation of protein corona. However, ANP functionalized NPs presented enhanced long-term colloidal stability, no hemotoxicity and retained cardiac cell viability. In vitro experiments confirmed that these NPs interacted preferentially with cardiomyocytes and non-myocytes through the binding of ANP to the natriuretic peptide receptors that are overexpressed in the infarcted region. Furthermore, ^111^In-labeled NPs were used for in vivo imaging experiments that again showed significant accumulation of the nanocarrier in the ischemic area and particularly in the endocardial layer of the left ventricle, underlining the targeting potential of ANP. At the final stage, the NPs were loaded with a hydrophobic small molecule (trisubsituted-3,4,5-isoxazole, C1) with cardioprotective properties. After intravenous administration of C1-NPs in a MI model in mice and in rats, a significant increase in the left ventricular ejection fraction and fractional shortening was observed. In addition, C1 decreased ERK1/2 phosphorylation, which is a key pathway leading to cardiac hypertrophy. In this way, the development of this nanocarrier provides a targeted therapeutic option against cardiac remodeling and further research should be conducted to study the effects of the delivery of other therapeutic molecules with this promising system [22].

The same group also developed a dual drug loaded dextran-based system for cellular reprogramming of cardiac fibroblasts into cardiac myocytes, for cardiac regeneration purposes after MI. The spermine acetalated dextran-based nanomaterial (AcDXSp) was prepared with an oil-in-water single emulsion method and was further functionalized with PEG and ANP to increase its stability and its specificity for cardiac cells, respectively. Two small hydrophobic molecules (CHIR99021 and SB431542) were incorporated to these NPs in a ratio of 1:2 in order to induce cardiomyocytes’ proliferation. SB431542 is a TGF-β inhibitor and CHIR99021 is a Wnt activator, both capable of increasing the efficiency of direct reprogramming of fibroblasts into cardiomyocytes. The final size of these NPs was below 400 nm, with a positive surface charge and minimal toxicity to cardiomyocytes. In addition, the pH responsive nature of this nanomaterial can provide a triggered release of the drugs in the cytoplasm of cardiac cells upon internalization into the acidic subcellular compartments. It was found that at pH = 5, the release was much faster than at pH = 7.4. Finally, in vitro administration of dual-loaded dextran NPs to rat cardiac cells indicated that CHIR99021, on the one hand, stabilized β-catenin, and SB431542, on the other hand, prevented the translocation of Smad3 to the nucleus of fibroblasts, both modulating different pathways for fibroblast reprogramming and increasing heart regeneration potential [61]. Recently, Torrieri and coworkers presented an advanced approach for this system, aiming for better targeting efficiency. Instead of using only ANP peptide for accumulation in cardiac cells, they also exploited linTTP1 peptide (AKRGARSTA) for targeting macrophages, associated with atherosclerotic plaques. This system was found to be safe for macrophages cell lines and primary macrophages, presenting no toxicity. In vitro experiments with macrophages indicated that the NPs could associate well with them and especially with M2-like macrophages, making them a possible target for efficient delivery. Interestingly, it was found that this system showed higher association versus uptake ratio towards M2-like macrophages and enhanced delivery of the drugs by dual targeting procedure [62].

Gold NPs are already known for their various applications in many fields of material science, as they have unique optical and physical properties. They are also an emerging type of DDS that can be easily functionalized and serve as vehicles for various therapeutic agents, as well as for imaging and diagnosis of diseases. Due to their potential biomedical applications, such as in photothermal therapy [63], many researchers are trying to develop gold-based nanosystems for curing different diseases. Recently, Somasuntharam and coworkers presented a very promising system for the treatment of inflammatory response caused during acute MI (Figure 5). They synthesized 80 nm sized (as determined by dynamic light scattering) gold NPs functionalized on their surface with deoxyribozyme (Dz) (Dz-AuNPs). DNAzymes are DNA oligonucleotides that are capable of inhibiting the activation of tumor necrosis factor α (TNF-α), a key factor implicated in many pathological processes, such as cellular inflammation, causing damage to the infarcted myocardium. Dz-AuNPs possess enhanced stability, cytocompatibility and in vitro studies showed that they are able to internalize in cardiomyocytes and in macrophages. Ex vivo fluorescent imaging revealed predominant accumulation in heart and secondary accumulation in liver and kidney. In vivo experiments in a rat model of acute MI exhibited a 50% reduction in TNF-α, followed by several beneficial effects, such as reduction in pro-inflammatory cytokines that regulate cell survival or apoptosis and trigger oxidative stress. Furthermore, echocardiography and pressure–volume catheter analysis followed by hemodynamic measurements showed a 40% improvement in cardiac function, while many cardiac indices, such as ejection fraction, end-diastolic/systolic pressure, end systolic volume, all improved. All these features indicate that Dz-AuNPs seem to be a potent vehicle for MI treatment by reducing local inflammatory responses and improving cardiac cell survival [64].

In a different study, Tian et al., reported the effect of bare 10 nm sized PEG-coated gold nanoparticles in MI in mice. The results showed that AuNPs accumulated in the heart, specifically after infarction, decreased infarct size, inhibited heart fibrosis by reducing gene expression of collagen I and collagen III, limited inflammation by TNF-α reduction and improved contractile function. However, administration of AuNPs had no effect on cardiac cell apoptosis and could not ameliorate cardiac hypertrophy post-MI [65].

An interesting work was referred by Ruvinov for the development of an injectable system consisting of an alginate biomaterial which could bind, with great affinity, to both insulin-like growth factor-1 (IGF-1) and hepatocyte growth factor (HGF). These factors have cytoprotective, anti-fibrotic and pro-angiogenic properties and have been utilized for infarct repair. With the use of this biomaterial, the two factors could be co-delivered to the injured area of the heart, leading to an efficient system for MI treatment. IGF-1 and HGF were reversibly bounded to alginate sulfate and both retained their bioactivity after bioconjugation, while at the same time they were protected from proteolysis in the MI environment. Release studies showed that the release of IGF-1 was quicker than the release of HGF, leading to sequential delivery. In vitro experiments revealed that after treatment with this double-loaded biomaterial, cardiomyocyte apoptosis was inhibited. The therapeutic potential of this dual IGF-1/HGF alginate biomaterial was proved in a rat model of acute MI after an intramyocardial injection. Specifically, four weeks after administration, scar thickness was increased, infarct expansion was prevented, scar fibrosis was decreased, vessel density increased and finally tissue restoration was achieved [66].

Hao’s group reported the synthesis of an alginate hydrogel to study the effect of delivery and sequential release of vascular endothelial growth factor-A_165_ (VEGF-A_165_) and platelet-derived growth factor-BB (PDGF-BB) to the heart after MI. These two factors both play a key role in angiogenesis. The hydrogel was prepared by combining alginate with two different molecular weights (5000 Da and 250,000 Da) and in vitro studies were conducted to monitor the release kinetics. It was found that VEGF-A_165_ released first, followed by a slower release of PDGF-BB. Both factors exhibited a sustained release for a 30-day period. Four weeks after intramyocardial administration, increased capillary densities were observed in periinfarct tissue sections and higher α-actin smooth muscle cell vessel density was noticed. In addition, the dual factor delivery increased the systolic velocity–time integral, a marker for proper cardiac function. In summary, single injection of this double loaded hydrogel induced angiogenesis and improved cardiac function in a rat model of MI and it was found to have enhanced therapeutic effects compared to single factor administration [67].

Recently, Tang and his team proposed for the first time the use of carbon nanocapsules (CNCs) as drug delivery systems for thromboembolism treatment in mice. Carbon nanocapsules were found to be biocompatible systems when injected intravenously, in contrast with carbon nanotubes that are toxic. Specifically, they developed heparin-conjugated CNCs (CNC-H), which exhibited high antithrombotic activity. Heparin is commonly used as an anticoagulant and has thrombolytic properties. It was conjugated to the surface of CNCs via EDC after acid modification of their surface. The diameter of CNC-H did not exceed 50 nm and drug loading was found to be circa 40%, as determined by TGA. Release experiments showed a burst release at the beginning, followed by a slower release rate. In vivo administration of CNC-H retarded thrombus formation, retained blood flow to acceptable levels and prolonged blood flow decrease [68].

Cell transplantation is an alternative method for the treatment of cardiac injury. However, the low cell survival and retention induced by this approach hampers its further application. Lu and coworkers have reported for the first time the preparation of a temperature-responsive chitosan hydrogel that protects and delivers embryonic stem cells (ESCs) to the damaged myocardium. Chitosan is a biocompatible and biodegradable material that is liquid at 4 °C and becomes solid at 37 °C within 15 min. This in situ gelation process makes chitosan a potent scaffold for the delivery of therapeutics to the area of interest. ESCs survived well when incorporated into the hydrogel in vitro. Four weeks after injection of the hydrogel into the infarct wall, graft size and cell retention increased. Echocardiography experiments exhibited a significant improvement in end-systolic/diastolic diameter, ejection fraction, and fractional shortening of the heart. Moreover, the infarct size was reduced by over 50%, the ventricular wall thickness was increased by over 50%, as compared with control groups, and the microvessel density was higher. Overall, this chitosan-based vehicle efficiently delivered stem cells to the affected area and considerably enhanced the heart function, making it a potential candidate for injured myocardium treatment [69].

A different method for cell replacement was followed by Ottersbach et al. They used embryonic cardiomyocytes (eCMs) and embryonic stem cell-derived cardiomyocytes (ES-CM) loaded with silica-coated magnetic NPs (SOMag5 MNPs) to increase their engraftment to the damaged myocardium. SOMag5 MNPs exhibited good loading and magnetization properties, while cytotoxic effects were absent (200 pg Fe/cell). With the application of magnetic field by a custom made 1.3 T magnet, at 5 mm distance and for 10 min, they achieved a 7-fold and 3.4-fold increase in engraftment of eCMs and ES-CM 2 weeks and 8 weeks after transplantation in rats. The MNPs were localized, mainly intra- and peri-cellularly, and accumulated in the heart, specifically to the left ventricle, even two months post-operation. The viability and the proliferation rate of the engrafted cells increased significantly, while better differentiation and an augmentation of cell to cell contacts was noticed. Furthermore, the left ventricle function strongly improved, as the left ventricular ejection fraction increased and end diastolic pressure and volume reduced. Overall, this novel nanosystem gives insight into a new cell transplantation approach for cardiomyoplasty [70]. A similar study was also presented by Zhang who labeled endothelial progenitor cells with silica-coated MNPs to restore cardiac function [71].

A short time ago, Miragoli and his team presented an unconventional approach for the delivery of a therapeutic peptide to the heart via inhalation. Treatment with inhalation delivery of drugs is common for respiratory diseases, although no such therapies have been found to treat cardiovascular diseases, until now. This team designed negatively charged calcium phosphate NPs (CaPs) that are bioresorbable, biocompatible and smaller than 50 nm. These particles, loaded with a cell-penetrating mimetic peptide (R7W-MP, DQRPDREAPRS), can be translocated from the pulmonary tree to the bloodstream and then to the heart. CaPs-R7W NPs can be internalized in cardiomyocytes by passing through their cell membrane, as in vitro experiments showed, and release their cargo. In vivo biodistribution studies revealed that NPs were rapidly deposited to the myocardium, as confirmed by fluorescence molecular tomography imaging and transmission electron microscopy. Notably, echocardiography showed that after administration of CaPs-R7W NPs in rats suffering from diabetic cardiomyopathy, complete restoration of heart contractility and function was achieved. These data underscore the importance of this DDS, as its non-invasive method and its efficacy open avenues for the treatment of CVDs [72].

## 4. Conclusions

The aim of this review is not only to provide an account of acquired knowledge on nanostructured materials developed for targeted drug delivery, but also to show that this concept can be easily applied to cardiovascular diseases. In the future, new revolutionized nanomaterials may play a key role in the establishment of innovative treatment for the challenge of current CVDs. All in all, it is clear that the application of nanotechnology to medicine is at the forefront of innovation for modern health care and represents a promising approach for efficient delivery of therapeutics.

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
