# Peer review of "Nanostructured Polymeric, Liposomal and Other Materials to Control the Drug Delivery for Cardiovascular Diseases"

_pharmaceutics, 2020, doi:10.3390/pharmaceutics12121160_

Round 1

Reviewer 1 Report

In this review Pharmaceutics-1014995) titled, “Nanostructured Polymeric, Liposomal and Other Materials to Control the Drug Delivery for Cardiovascular Diseases”, the authors discuss the recent developments on using biocompatible nanomaterials as potential therapeutic interventions for Cardiovascular diseases.

The review is current and the references used are up to date. Surely the topic falls within the objectives of this journal  and is therefore publishable after revision of the following aspects:

 Key points that need addressing

  1. Please insert a chapter dedicated to biocompatible materials reporting the desired characteristics for cardiac use.
  2. Cardiac Targeting Strategies: in this chapter the authors advocate active and passive drug targeting, but this discussion is not actually focused on heart tissue and appears rather limited to solid tumors. Therefore, the same authors underline that this topic is controversial and therefore they could better argue it, inserting the most innovative strategies that come closest to the targeted delivery of a drug on heart tissue. For example, the concepts of responsive nanoparticles, double prodrug, AuNps or multifunctional silica NPs, discussed below, could be introduced here. As well as, the approach of smart polymer and hydrogels.
  3. Polymeric Nanoparticles: in this chapter, the authors define polymeric nanoparticles as micelles consisting of amphiphilic polymers obtained by self-assembly. The definition of polymeric nanoparticles is instead broader and the methods of preparation used are numerous because they take into account the different chemical-physical properties of the drugs and polymers used. Thus, for example, PLGA is a hydrophobic polyester, while its copolymer with PEG is usually an amphiphilic block system. PLGA forms biodegradable nanospheres, suitable for hydrophobic drug incorporation and / or sustained release, although it has also been extensively explored for hydrophilic drugs, while its PLGA-PEG copolymer forms core-schell micelles or stealth nanoparticles.The authors should therefore better argue this first part of chapter (lines 279 to 294) by highlighting the most sought-after features for release into cardiac tissue, so the reader can better follow the examples discussed below.

Author Response

We would like to thank the reviewers for their constructive comments that improved significantly our manuscript. Please find below the point-to-point answers to Reviewer #1

  1. We inserted two additional paragraphs on Chapter 2, i.e. the last two paragraphs of page 2, presenting the desired characteristics of the nanoparticles for cardiac use.
  2. We inserted the concepts of other types of nanoparticles used for heart diseases in Section 2. The functionality of these nanoparticles has been presented thoroughly in section 3. We added more information about targeting strategies specifically for heart diseases on section 2, indicating clearly the differences between tumor and heart tissue under stress. It is on page 3 lines 168-180 and 189-199.
  3. We agree and we added other types of polymeric nanoparticles on page 8 lines 499-503. We gave more general types of nanocostructs that can deliver drugs to pathological sites. We highlighted the features of the nanoparticles required for targeted heart delivery on section 2, page 2 the last two paragraphs and page 3 lines 169-181.

We would like to thank again Reviewer #1 for his consideration.

Reviewer 2 Report

In this paper, Dimitrios Skourtis et.al tried to give a review of the works on drug delivery for Cardiovascular Diseases. The authors give the background about the Cardiovascular Diseases. They especially collect many works on liposome and polymeric nanoparticles. It is interesting in general and can possibly serve as a good overview of this field.    However, I would suggest the authors revise their manuscript before considering the recommendation in the next step.

  • In the introduction, the authors put too many things to introduce Cardiovascular Diseases before drug delivery. For example, it might be better to put the paragraphs before line 74 into one or two paragraphs to give an overview and shortcomings of other treatments.
  • When introducing the passive targeting strategy, the authors introduce the EPR effect. The EPR effect is one of the most controversial mechanisms in the drug delivery process. The authors need to introduce the EPR effect more cautiously. Otherwise, it is highly possible to mislead readers. For instance, the authors might need to include Warren’s recent works about the EPR, where Warren challenged the EPR effect.
  • When introducing the specific works on the liposome, it is better to combine most of the detailed mentioned works into one complex figure. It will be much better to give the readers more information. Right now, the authors only include one figure for a single specific work. The same situation is applied to polymeric works. It is important as a reviewer works, it is better to reorganize the refereed works’ figures into one complex figure and put them into your own story.
  • It might be better to include some theoretical works to talk about the mechanisms, such as the cellular uptake process. For instance, the computational works on the endocytosis of liposomes in the group of Dr. Ying Li.
  • Finally, it is better to discuss the difficulty that the drug delivery field needs to overcome. As we know, the idea and lab experiments of nanomedicine are very promising. While there are only a few cases that can be applied in the clinic application.  

Author Response

We would like to thank the reviewers for their constructive comments that improved significantly our manuscript. Please find below the point-to-point answers to Reviewer #2

  • We reconstructed sections 2 and 3 in one entitled “Cardiac Targeting Strategies and Nanomedicine”. We removed completely some paragraphs to shorten the text and give an overview and shortcomings of cancer treatment.
  • We agree with the reviewer. We added the text in lines 168-179 highlighting the EPR effect more cautiously. We included the work of Warren et al. which challenged the EPR effect (reference 19).
  • We included one new Scheme, Scheme 1 to denote the functionality of polymeric and liposomal nanoparticles to treat heart diseases. In order to summarize all the works on liposomes and polymeric NPs we introduced two Tables with all the works referred. We think that this will help the readers to find a certain work more easily.
  • We included the work of Li et al. on the bottom of page 3 and the top of page 4. We included the corresponding references 25-27.
  • We described clearly the biological barriers that the NPs have to bypass particularly in heart diseases and we described the differences with other diseases like cancer in the bottom of page 2 and the top of page 3.

We would like to thank again Reviewer #2 for his consideration.

Round 2

Reviewer 1 Report

The authors addressed all responses. The manuscript has significantly improved and is ready for publicThe authors addressed all responses. The manuscript has significantly improved and is ready for publicationtion

Reviewer 2 Report

The author answered my questions.